# Effects of Artificial Light at Night on Photosynthesis and Respiration of Two Urban Vascular Plants

Yaxi Wei [1,2], Jiaolong Zhang [1,2], Dan Hu [1], Jian Zhang [3,4] and Zhen Li [3,4,*]

1   State Key Laboratory of Urban and Regional Ecology, Research Center for Eco-Environmental Sciences, Chinese Academy of Sciences, Beijing 100085, China
2   University of Chinese Academy of Sciences, Beijing 100049, China
3   State Key Laboratory of Simulation and Regulation of Water Cycle in River Basin, China Institute of Water Resources and Hydropower Research, Beijing 100038, China
4   Department of Water Ecology and Environment, China Institute of Water Resources and Hydropower Research, Beijing 100038, China
*   Correspondence: lizhen@iwhr.com

**Abstract:** The widespread use of artificial light at night (ALAN) due to urbanization and infrastructure development has raised concerns about its potential impacts on plant physiology. To explore the effects of ALAN with different light intensities on the photosynthesis and respiration of two urban vascular plants, *Euonymus japonicus* Thunb and *Rosa hybrida* E.H.L.Krause, under continuous and non-continuous-light conditions, respectively, a field experiment was conducted. Our findings indicate that continuous ALAN significantly inhibited the photosynthesis and respiration of the two plants, disrupting the carbon-balance pattern of their leaves during the day, but this effect is influenced by the intensity of ALAN and plant species. *Euonymus japonicus* Thunb is more susceptible to continuous ALAN than *Rosa hybrida* E.H.L.Kraus. Non-continuous ALAN did not significantly affect the photosynthesis and respiration of two species during the day. Furthermore, we observed that high light intensity at night could also impact the value of the net photosynthetic rate (Pn) of *Euonymus japonicus* Thunb during the night. Turning off light sources at night, reducing the light intensity, and cultivating ALAN-tolerant plants are effective measures to reduce the negative effects of ALAN on plants and maintain plants' normal carbon-balance mode. Future studies should explore the effects of different types of artificial-light sources combined with other environmental conditions on the photosynthesis and respiration of plants at the canopy scale.

**Keywords:** artificial light at night; photosynthesis; respiration; carbon balance; vascular plants





## 1. Introduction

Since the early 20th century, accompanied rapid urbanization and infrastructural development, the widespread utilization of artificial light at night (ALAN) has occurred worldwide [1]. While ALAN provides undeniable convenience to human society, it disrupts the natural patterns of light, resulting in environments located tens to hundreds of kilometers away from their natural brightness and contributing to the formation of light pollution [1,2]. Current estimates suggest that over 20% of the earth's land area is affected by light pollution, and the area exposed to such pollution is increasing [2,3]. Consequently, ALAN has emerged as a prominent human-driven factor that significantly endangers global biodiversity and ecosystem services [1,4,5]. A lot of research have been conducted to investigate the effects of ALAN on animals in the past few decades, and it has been proven that ALAN can affect the reproduction, distribution, and migration of sea turtles, birds, and moths [6–8].

Light is one of the essential environmental abiotic factors in the regulation of plant physiology and ecology [1,9]. As a resource, light can be captured by plants for photosynthesis. While the intensity of ALAN is generally lower than daytime solar radiation, there is a theoretical possibility that ALAN, particularly in cave systems, could provide sufficient

light to trigger minimal photosynthesis [10,11]. Therefore, the carbon-assimilation process in leaves under ALAN may still be neglected. Moreover, as a source of information, the intensity, duration, and spectral distribution of light have a great influence on plants in perceiving diurnal cycles, seasonal changes, and the characteristics of their surrounding environment [9,10]. This information, in turn, enables plants to regulate their physiological activities accordingly. However, the perception of ALAN by photoreceptors can disrupt information flow and interfere with the normal signaling and functions of these photoreceptors, ultimately impacting the daytime activities of plants [10]. Studies have shown that uninterrupted darkness is essential for plants to repair damage incurred during the day and for maintaining regular growth rhythms [9,10]. Vollsnes et al. [12] demonstrated that even weak night light can impede the nocturnal repair process of ozone-damaged leaves. Additionally, Gerrish et al. [13] established that several hours of darkness are necessary to repair DNA damage caused by UV-B radiation. Cathey and Campbell [14] pointed out that cutting off light sources for two to four hours at night will maintain the plant's natural rhythm. Nevertheless, ALAN, which encroaches on dark refuges in space, in time, and across wavelengths, weakens or even completely deprives the role of darkness in plant growth, affecting the normal physiological function of plants [10].

ALAN has been shown to exert significant effects on the photosynthesis and respiration of lower plants. Levy et al. [5] reported that ALAN decreases the photosynthetic rate and inhibits the respiration of algae that live in symbiosis with coral reefs. Ayalon et al. [4] revealed that symbiotic algae exposed to ALAN experience oxidative stress conditions and exhibit a lower photosynthesis performance, measured by electron transport rate, as well as changes in chlorophyll- and algae-density parameters. Additionally, Poulin et al. [11] pointed out that ALAN would adversely affect the photosynthesis of *Microcystis aeruginosa.*

The impacts of manually supplemented light at night on vascular plants in agriculture have been extensively investigated. Plant scientists and horticulturists have observed increased growth, daily carbon gain, and productivity in tomato plants with the use of night-time light [9,15,16]. However, detrimental effects have also been identified in some crops, such as a reduced nitrate content, maximum photosynthetic capacity, Rubisco activity, and electron transport, and an increased vascular-pattern complexity [9,16]. Recently, researchers have explored the effects of night-time light on plant physiology in non-agricultural fields. Zhang et al. [17] found that continuous ALAN can reduce chlorophyll content, increase malondialdehyde content, and inhibit growth in ryegrass, but an appropriate extension of the photoperiod can increase its biomass. Lo Piccolo et al. [18] found that the photosynthetic rates of *Platanus* × *acerifolia* and *Tilia platyphyllos* decreased significantly under continuous ALAN. Meravi and Prajapati [19] studied seven species of vascular plants under street lights and found that the leaves of vascular plants near the street light showed a decreased maximum photochemical quantum yield of photosystem II and a photochemical quantum yield of photosystem II. Overall, existing studies on the relationship between night-time light and vascular plants have primarily focused on relatively high light intensities, exceeding the plants' light-compensation point. Urban areas are heavily impacted by ALAN, and vascular plants play a crucial role in urban ecosystems. In urban areas, except for the parts of the plants near the light source, the intensity of the ALAN received by the plants was basically below the compensation point. At present, understanding of the effects of ALAN with intensities below the light-compensation point on plant photosynthesis at night remains limited. Many scholars have found that low-light environments can inhibit the dark-respiration rate of plant leaves [20]. However, few studies have investigated the effects of ALAN on respiration in vascular plants [1,9,18]. Whether ALAN can affect leaf respiration and the subsequent net accumulation of organic matter in plants still needs to be further discussed.

Furthermore, urban environments exhibit significant heterogeneity in night-time light conditions, with some habitats experiencing continuous night-time illumination while others experience non-continuous light exposure. Plant photosynthesis and respiration are fundamental physiological processes that directly influence the growth and carbon balance

of plants. In this study, the typical urban vascular plants *Euonymus japonicus* Thunb and *Rosa hybrida* E.H.L.Krause, which are widely used or grown for urban greening in China, were taken as the research objects to investigate the effects of ALAN with different light intensities on their photosynthesis and respiration processes under continuous ALAN and non-continuous ALAN.

## 2. Materials and Methods

### 2.1. Experimental Design

A field-manipulation experiment was conducted in Yanqing, located in the northwest of Beijing, China (40°29′ N, 115°59′ E). In August 2021, a designated experimental area with a size of 36 × 15 m was selected and divided into several 2 × 2 m plots, with a 1.5 m buffer zone between them. Then, 21 plots were selected and randomly assigned to seven treatment groups, with three replicates per treatment. A 2 m long wooden frame positioned 1.7 m above the ground was installed at the center of each plot to accommodate the placement of the light sources. One group served as the control without any light source, while the other six groups were divided based on the time and intensity of light exposure. Two light durations were set, continuous night-time light and non-continuous night-time light, and there were low, middle, and high light intensities for each duration. The duration of light sources' irradiation was regulated by a light-sensitive switch. All light sources were turned on at dusk, and the continuous night-time light was turned off at dawn the next day, while the non-continuous light was turned off at 23:00. The light intensities were achieved using LED lights of 3 W, 5 W, and 9 W (OUPU China, CCT6500K), respectively, and the illumination was adjusted by partial occlusion with a black shading net [21], but the spectral distribution was not changed (Figure S1). The mean light intensities of the low, middle, and high factors measured using the illuminometer (XIMA China) on the ground of the plots were 22 lux, 57 lux, and 94 lux, respectively. Lampshades and black plastic plates were used to shade the light source within each plot, preventing interference with other treatments. (Images of the field experiment are shown in Figure S2)

Also in August 2021, two-year-old *Euonymus japonicus* Thunb and *Rosa hybrida* E.H.L.Krause plants with mean heights of 64.63 cm and 65.18 cm, respectively, and basal diameters of 12.28 mm and 11.93 mm, respectively, were purchased from a plant nursery. Finally, a total of 126 plants of each species were randomly divided equally among the 21 plots, and planted directly into the soil. During the whole experiment, the plants received sufficient irrigation and fertilization, while concurrently undergoing a regular eradication of weeds and preventative measures against pests and diseases. Pruning and antifreeze measures were applied in winter to ensure their survival.

In summary, the experiment involved two plant species, *Euonymus japonicus* Thunb and *Rosa hybrida* E.H.L.Krause, and seven treatments, No ALAN (CK), C-L (continuous ALAN with low light intensity), C-M(continuous ALAN with middling light intensity), C-H (continuous ALAN with high light intensity), NC-L (non-continuous ALAN with low light intensity), NC-M (non-continuous ALAN with middling light intensity), and NC-H (non-continuous ALAN with high light intensity). Each treatment included three replicated plots, with each plot containing six plants. Considering that July represents the typical summer period, which is the flourishing period of the growth of the two plants, and September represents the typical autumn period, which is the beginning of the decline period of the growth of the two plants, we chose July and September to measure the relevant indicators of the two plants.

### 2.2. Measurements of the Net Photosynthetic Rate (Pn) Responses to Photosynthetic Photon Flux Density (PPFD) and $CO_2$ Concentration

The responses of the Pn to PPFD and $CO_2$ concentration were measured using a portable photosynthesis analysis system (LI-6400XT, LI-COR, Lincoln, NE, USA) on sunny windless days from 08:30 to 15:30 in July and September 2022, respectively. For either of the two species in each plot, two fully expanded leaves from the upper part of two plants,

which were close to the center of the plot, were measured. During the measurements, the air flow rate was 500 $\mu mol \cdot s^{-1}$, the relative humidity (RH) was set to 44%–60%, and the temperature of the leaf chambers were set to 28 °C in July and 25 °C in September.

In the measurement of the Pn responses to PPFD, the reference $CO_2$ concentration was controlled at 400 $\mu mol \; mol^{-1}$ by using a small $CO_2$ cylinder. Before the determination, the leaves were induced under saturated light intensity until the photosynthetic parameters were stable, and then the Pn was determined automatically under the PPFD gradients of 1800, 1600, 1400, 1200, 1000, 800, 600, 400, 200, 100.50, 30, 10, 5, and 0 $\mu mol \; m^{-2}s^{-1}$ (PPFD of *Euonymus japonicus* Thunb from 1600 $\mu mol \; m^{-2}s^{-1}$ to start).

In the measurement of the Pn responses to $CO_2$ concentration, the PPFD was set to 1200 $\mu mol \; m^{-2}s^{-1}$, which is close to the saturation light intensity of both plants. Before the determination, the leaves were induced under this light intensity until the photosynthetic parameters were stable, and then the Pn was determined automatically at $CO_2$ concentration levels of 400, 300, 200, 150, 100, 50, 400, 600, 800, 1000, 1200, and 1500 $\mu mol \; mol^{-1}$. $CO_2$ concentration was controlled using a small $CO_2$ cylinder.

Several studies have indicated that the fitting model applicable to the response of Pn to PPFD and $CO_2$ concentration varies across different adversity stress conditions [22]. The exponential model, as described by Bassman and Zwier [23], was employed to fit the Pn responses to PPFD in order to determine the maximum net photosynthetic rate ($Pn_{max}$, $\mu mol \; CO_2 \cdot m^{-2}s^{-1}$), light-compensation point (lc, $\mu mol \; photon \cdot mol^{-1}$), and dark-respiration rate (Rd, $\mu mol \; CO_2 \cdot m^{-2}s^{-1}$). The modified rectangular hyperbolic model, as described by Ye and Yu [24], was utilized to fit the Pn responses to $CO_2$ in order to obtain the photosynthetic capacity ($A_{max}$, $\mu mol \; CO_2 \cdot m^{-2}s^{-1}$), photorespiration rate (Rp, $\mu mol \; CO_2 \cdot m^{-2}s^{-1}$), $CO_2$-compensation point ($\Gamma$, $\mu mol \; CO_2 \cdot mol^{-1}$), and saturated intercellular $CO_2$ concentration ($Ci_{sat}$, $\mu mol \; CO_2 \cdot mol^{-1}$). Additionally, the biochemical model, as described by Farquhar et al., [25] was employed to estimate the maximum carboxylation rate ($Vc_{max}$, $\mu mol \; CO_2 \cdot m^{-2}s^{-1}$), maximum electron-transfer rate ($J_{max}$, $\mu mol \; CO_2 \cdot m^{-2}s^{-1}$), and triose phosphate-utilization rate (TPU, $\mu mol \; CO_2 \cdot m^{-2}s^{-1}$) from the $CO_2$ photosynthetic response.

### 2.3. Measurements of Chlorophyll Fluorescence Parameters

For either of two species in each plot, two mature leaves at the top from two different plants which are close to the center of the plot were chosen in July and September 2022, respectively. Chlorophyll fluorescence parameters were measured using a portable photosynthesis analyzer (LI-6400XT, LI-COR, Lincoln, NE, USA) on sunny windless mornings from 8:30 to 11:00. During the measurements, photosynthetically active radiation was 1200 $\mu mol \; m^{-2}s^{-1}$. When the instrument was stable, the following parameters were determined: effective quantum yield ($\Phi_{PSII}$), apparent electron-transfer rate (ETR), and photochemical quenching parameter ($q_p$).

### 2.4. Measurement of $CO_2$ Exchange at Night

For either of the two species in each plot, one mature leaf at the top from plant nearest to the light source was chosen in July and September 2022, respectively. The Pn at night was measured using a portable photosynthesis analyzer (LI-6400XT, LI-COR, Lincoln, NE, USA) on cloudless nights from 21:00 to 22:30 and 2:00 to 3:30 the next day in ambient an environment, respectively.

### 2.5. Statistical Analysis

We analyzed the differences of the indicators under different night light intensities and the influence of the interaction between night light intensity and season on some indicators under continuous-light conditions and non-continuous-light conditions, respectively. The differences between different treatments were analyzed through one-way variance followed by an LSD test, and the interaction between light intensity and season was analyzed through two-factor variance. Normal and homogeneous tests were performed in the data analysis. All data analyses were made using SPSS software v22 and were plotted with Origin software 2023.

## 3. Result

### 3.1. Pn Response to PPFD

Compared with the control, *Euonymus japonicus* Thunb exhibited significant decreases in $Pn_{max}$ and Rd and a significant increase in Rd/Ag under continuous ALAN in both July and September (Figure 1a,b,d). $Pn_{max}$ significantly decreased by 10% (11%), 11% (17%), and 18% (20%) in July (September) under low, medium, and high light intensities, respectively. Rd significantly decreased by 2% (2%), 3% (2%), and 3% (3%) in July (September) under low, medium, and high light intensities, respectively. Rd/Ag significantly increased by 9% (10%), 8% (16%), and 17% (20%), respectively in July (September) under low, medium, and high light intensities, respectively. In July, $Pn_{max}$ under a high light intensity was significantly lower than that under low and medium light intensities, Rd/Ag under a high light intensity was significantly higher than that under low and medium light intensities, and $Pn_{max}$ and Rd/Ag under low and medium light intensities were not significantly different. In July, Rd was significantly lower under a high light intensity compared to a low light intensity, but no significant difference was observed between medium light intensity and low/high light intensity. In September, $Pn_{max}$ and Rd under a high light intensity were significantly lower than those under a low light intensity, Rd/Ag under a high light intensity was significantly higher than that under a low light intensity, and $Pn_{max}$, Rd, and Rd/Ag under a medium light intensity did not show significant differences compared to high and low light intensities. For *Rosa hybrida* E.H.L.Krause, significant differences in $Pn_{max}$, Rd, and Rd/Ag were observed only under a continuous high light intensity compared to the control in both July and September (Figure 1e,f,h). $Pn_{max}$ and Rd decreased by 12% (19%) and 2% (3%) in July (September), respectively. Rd/Ag increased by 10% in July and 18% in September. Compared to the control, continuous ALAN did not significantly affect the lc of *Euonymus japonicus* Thunb and *Rosa hybrida* E.H.L.Krause in July; however, a high light intensity significantly increased the lc of both species in September (Figure 1c,g). In both July and September, there were no significant differences in $Pn_{max}$, Rd, lc, and Rd/Ag among treatments under non-continuous ALAN for both *Euonymus japonicus* Thunb and *Rosa hybrida* E.H.L.Krause (Figure 2).

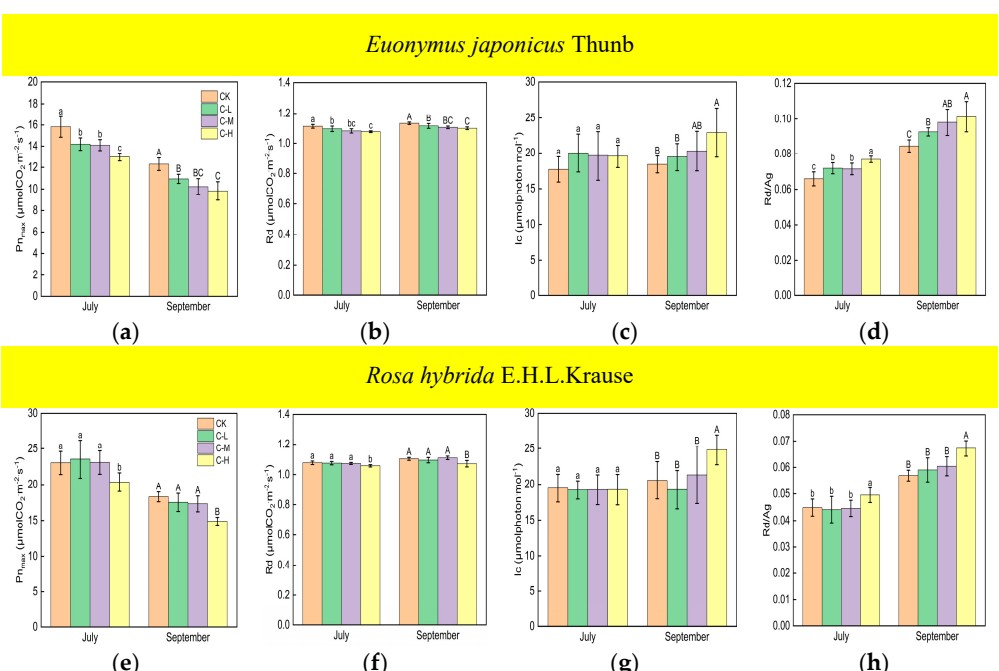

**Figure 1.** Effects of light intensity on $Pn_{max}$, Rd, lc, and Rd/Ag (Ag = $Pn_{max}$ + Rd) of *Euonymus japonicus* Thunb (**a–d**) and *Rosa hybrida* E.H.L.Krause (**e–h**) under continuous ALAN. Different lowercase and capital letters within the columns indicate significant differences among the treatments at *p* < 0.05 in July and September, respectively. Data are mean values ± SDs (*n* = 6).

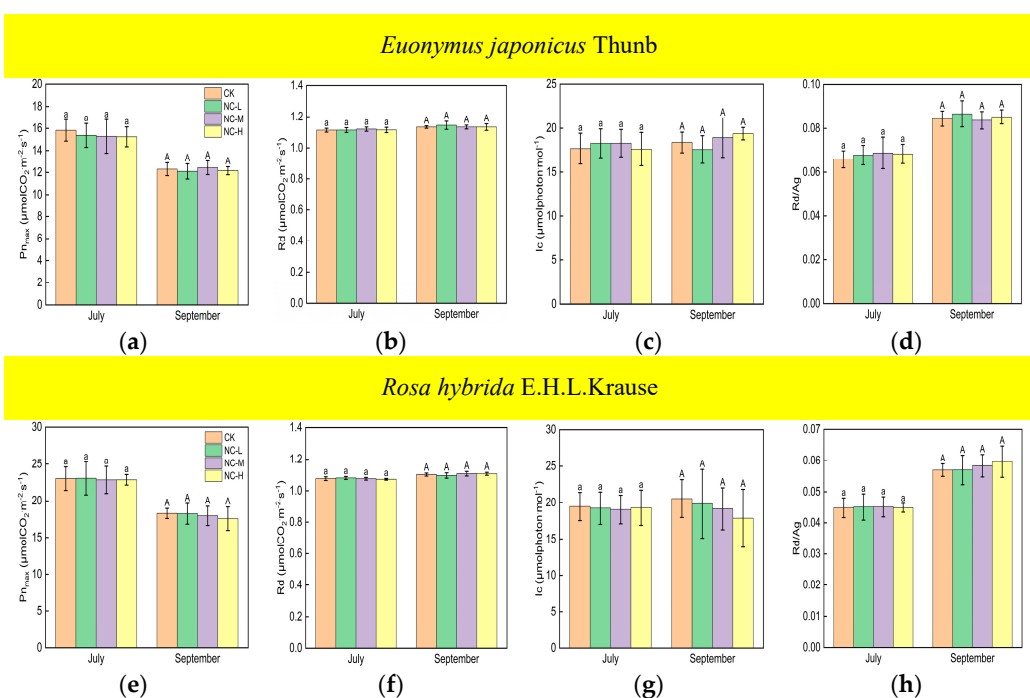

**Figure 2.** Effects of light intensity on $Pn_{max}$, Rd, lc, and Rd/Ag of *Euonymus japonicus* Thunb (**a**–**d**) and *Rosa hybrida* E.H.L.Krause (**e**–**h**) under non-continuous ALAN. Different lowercase and capital letters within the columns indicate significant differences among the treatments at $p < 0.05$ in July and September, respectively. Data are mean values $\pm$ SDs ($n = 6$).

### 3.2. Pn Response to $CO_2$ Concentration

Under continuous ALAN, the $A_{max}$, $Vc_{max}$, $J_{max}$, and TPU of *Euonymus japonicus* Thunb showed significant decreases compared to the control group in both July and September (Figure 3a,e–g). $A_{max}$ decreased by 6% (8%), 7% (11%), and 7% (16%) in July (September) under low, medium, and high light intensities, respectively. $Vc_{max}$ decreased by 9% (12%), 9% (13%), and 9% (18%) in July (September) under low, medium, and high light intensities, respectively. $J_{max}$ decreased by 11% (12%), 11% (13%), and 12% (17%) in July (September) under low, medium, and high light intensities, respectively. TPU decreased by 6% (11%), 6% (15%), and 7% (19%) in July (September) under low, medium, and high light intensities, respectively. In July, no significant differences were found in $A_{max}$, $Vc_{max}$, $J_{max}$, and TPU in the low-, medium-, and high-light-intensity treatments under continuous ALAN. In September, $A_{max}$, $Vc_{max}$, and TPU under a high light intensity were significantly lower than those under a low light intensity, and $A_{max}$, $Vc_{max}$, and TPU under a medium light intensity did not significantly differ from those under low and high light intensities; moreover, there were no significant differences in $J_{max}$ in the three light-intensity treatments. Compared with the control, the $A_{max}$, $Vc_{max}$, $J_{max}$, and TPU of *Rosa hybrida* E.H.L.Krause differed significantly only at a continuous high light intensity (Figure 3h,l–n). $A_{max}$, $Vc_{max}$, $J_{max}$, and TPU were significantly reduced by 13% (14%), 14% (16%), 15% (17%), and 14% (13%) in July (September). Continuous ALAN did not significantly affect the Rp, $\Gamma$, or $Ci_{sat}$ of the *Euonymus japonicus* Thunb and *Rosa hybrida* E.H.L.Krause in both July and September (Figure 3b–d,i–k). For both species, $A_{max}$, Rp, $\Gamma$, $Ci_{sat}$, $Vc_{max}$, $J_{max}$, and TPU did not differ significantly in the treatments under non-continuous ALAN in both July and September (Figure 4).

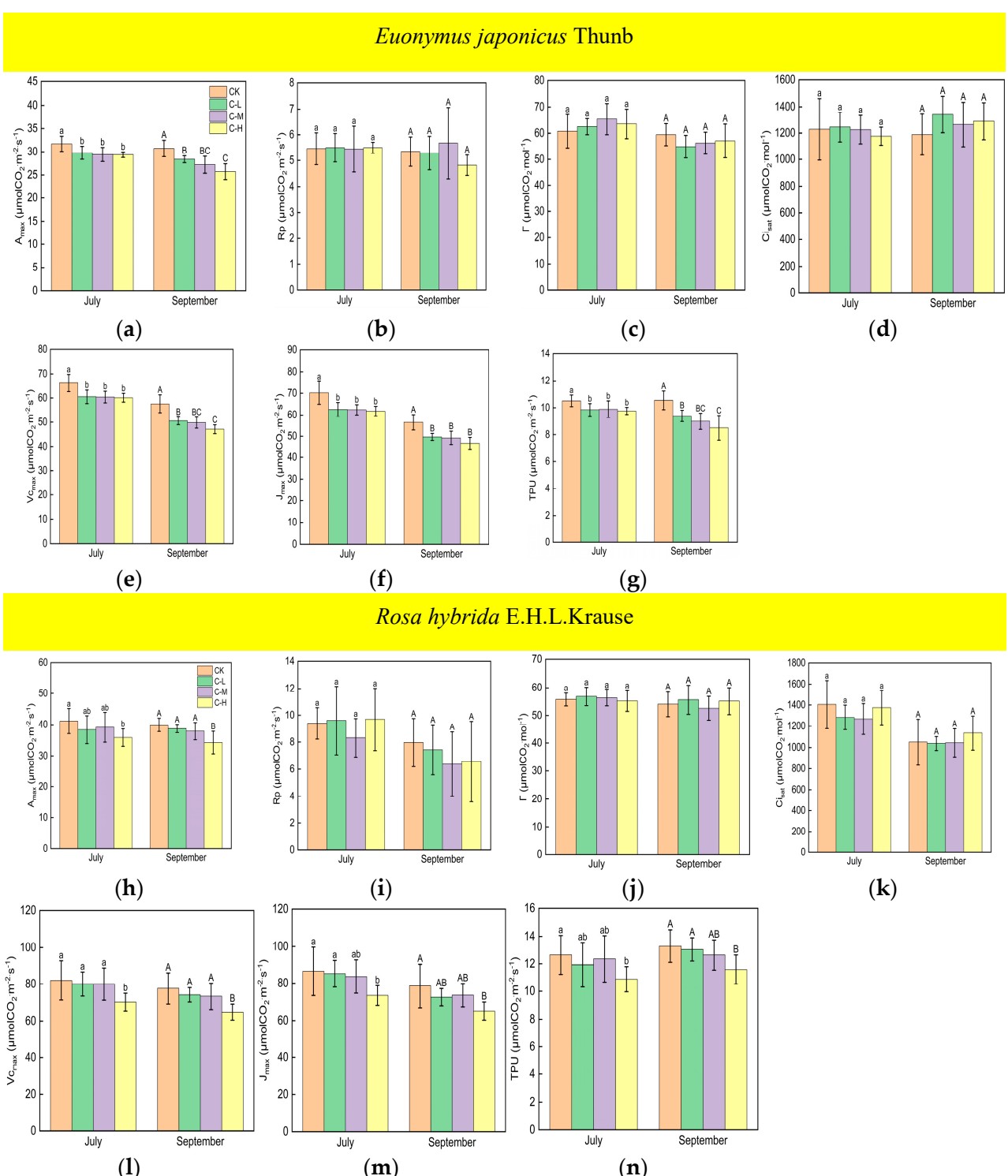

**Figure 3.** Effects of light intensity on the $A_{max}$, Rp, $\Gamma$, $Ci_{sat}$, $Vc_{max}$, $J_{max}$, and TPU of *Euonymus japonicus* Thunb (**a–g**) and *Rosa hybrida* E.H.L.Krause (**h–n**) under continuous ALAN. Different lowercase and capital letters within the columns indicate significant differences among the treatments at $p < 0.05$ in July and September, respectively. Data are mean values ± SDs ($n = 6$).

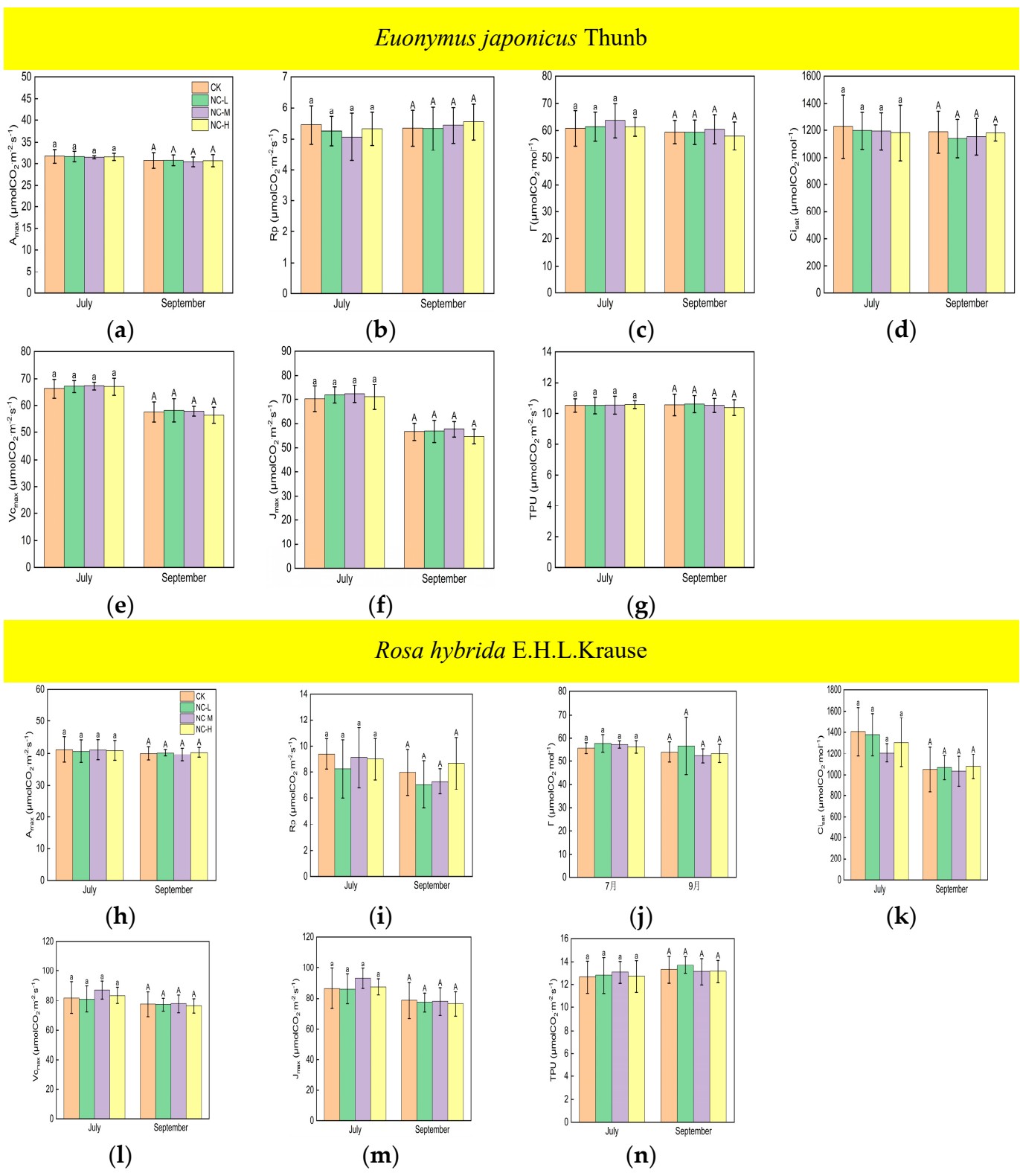

**Figure 4.** Effects of light intensity on $A_{max}$, Rp, $\Gamma$, $C_{isat}$, $Vc_{max}$, $J_{max}$, and TPU of *Euonymus japonicus* Thunb (**a–g**) and *Rosa hybrida* E.H.L.Krause (**h–n**) under non-continuous ALAN. Different lowercase and capital letters within the columns indicate significant differences among the treatments at $p < 0.05$ in July and September, respectively. Data are mean values $\pm$ SDs ($n = 6$).

*3.3. Fluorescence Parameters*

The values of the ETR, $\Phi_{PSII}$, and $q_p$ of *Euonymus japonicus* Thunb showed significant reductions under continuous ALAN compared to the control in both July and September (Figure 5a–c). ETR decreased by 14% (10%), 16% (12%), and 19% (18%) in July (September) under low, medium, and high intensities, respectively. $\Phi_{PSII}$ decreased by 14% (10%), 16% (12%), and 19% (19%), respectively, in July (September) under low, medium, and high intensities, respectively. $q_p$ decreased by 17% (17%), 18% (14%), and 17% (20%) in July (September) under low, medium, and high intensities, respectively. There was no significant difference in ETR, $\Phi_{PSII}$, and $q_p$ under different light intensities in July. However, in September, the ETR and $\Phi_{PSII}$ under a high light intensity were significantly lower than those under a low light intensity, while the ETR and $\Phi_{PSII}$ under a medium light intensity did not show significant differences compared to high and low light intensities. For *Rosa hybrida* E.H.L.Krause, ETR, $\Phi_{PSII}$, and $q_p$ decreased significantly only under a continuous high light intensity in both July and September (Figure 5d–f). ETR, $\Phi_{PSII}$, and $q_p$ decreased by 10% (15%), 10% (15%), and 8% (12%), respectively, in July (September). In both July and September, there were no significant differences in ETR, $\Phi_{PSII}$, and $q_p$ in the treatments under non-continuous ALAN for both *Euonymus japonicus* Thunb and *Rosa hybrida* E.H.L.Krause (Figure 6).

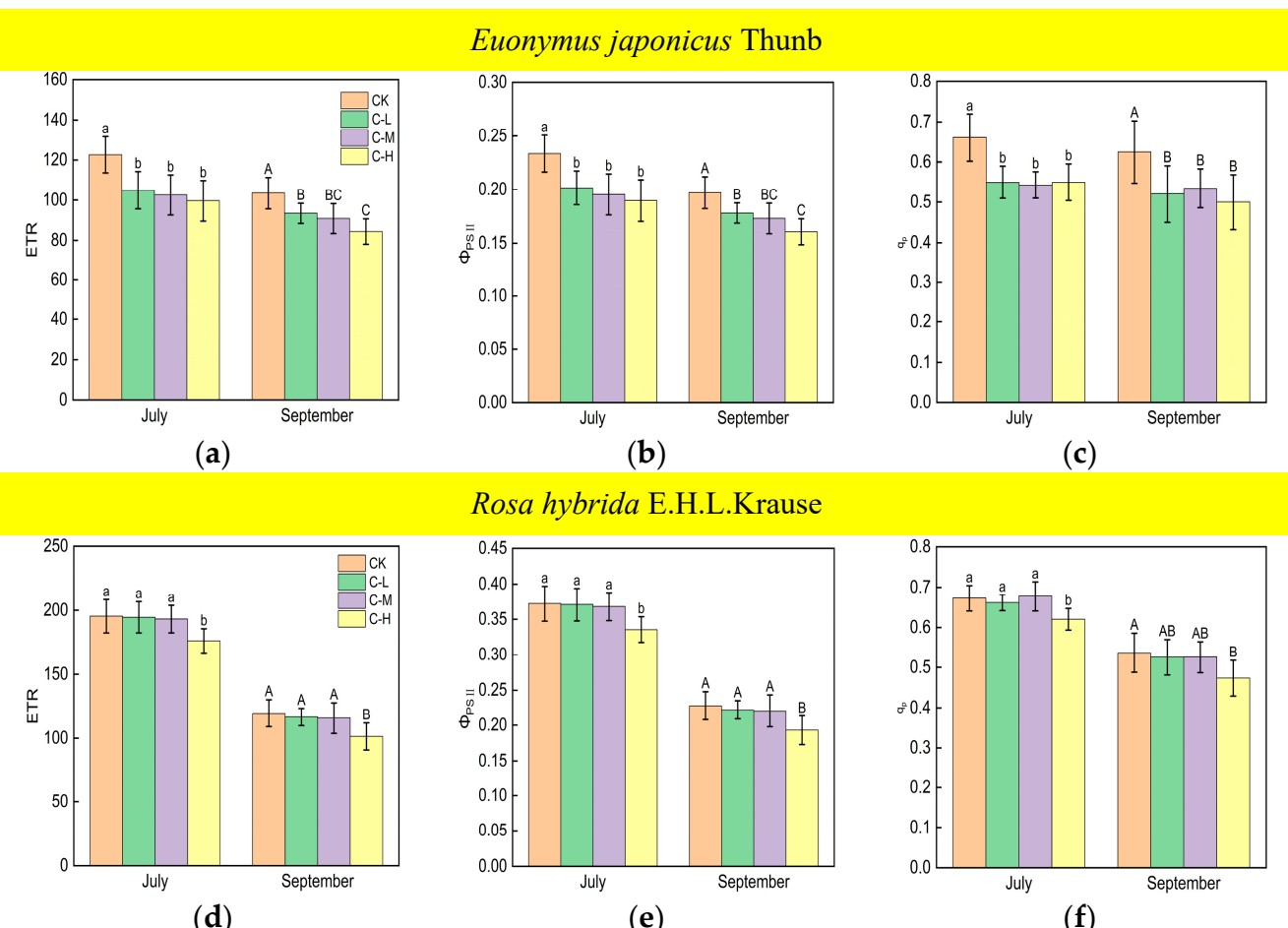

**Figure 5.** Effects of light intensity on the ETR, $\Phi_{PSII}$, and $q_p$ of *Euonymus japonicus* Thunb (**a–c**) and *Rosa hybrida* E.H.L.Krause (**d–f**) under continuous ALAN. Different lowercase and capital letters within the columns indicate significant differences among the treatments at $p < 0.05$ in July and September, respectively. Data are mean values ± SDs ($n = 6$).

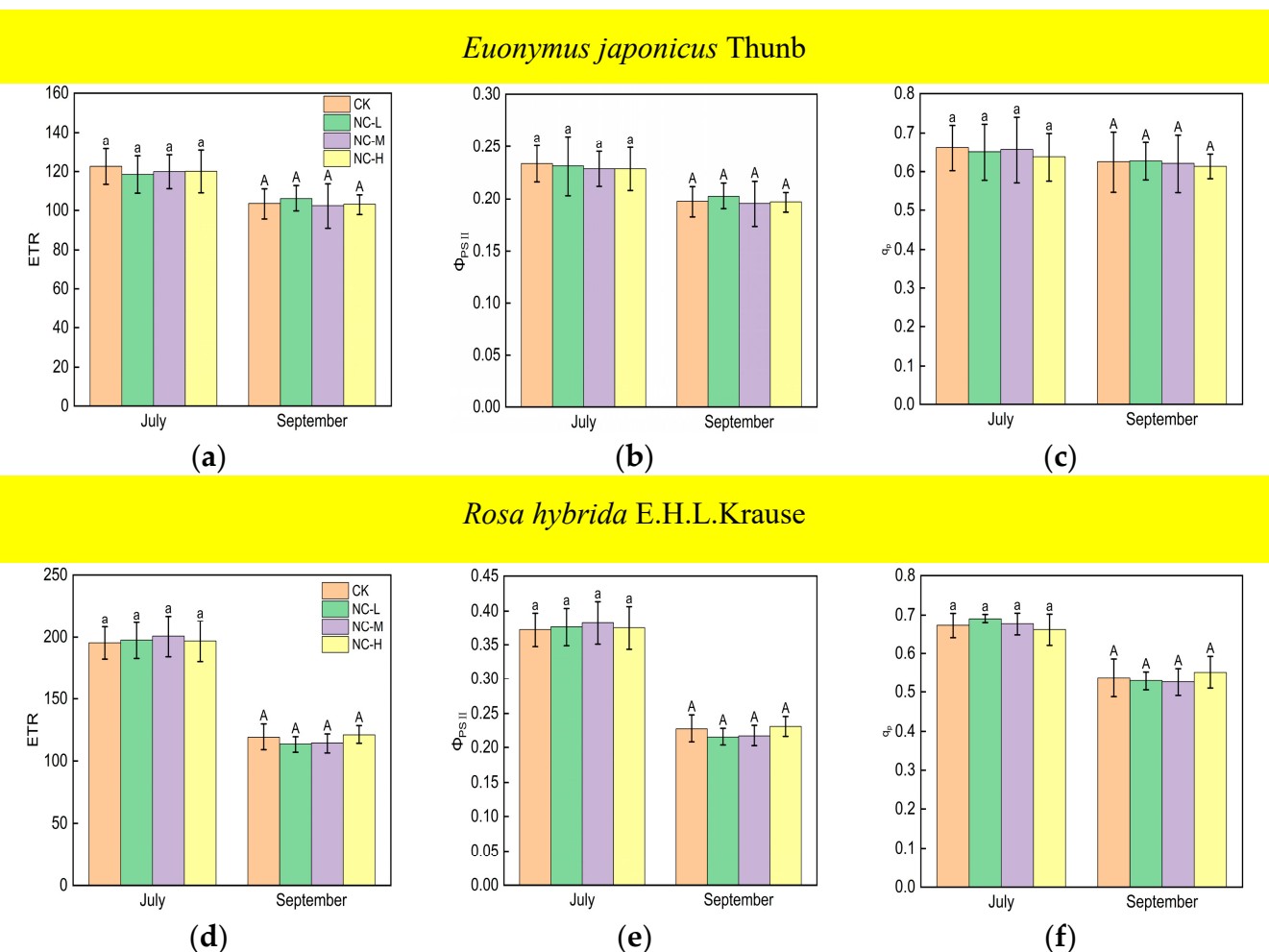

**Figure 6.** Effects of light intensity on the ETR, $\Phi_{PSII}$, and $q_p$ of *Euonymus japonicus* Thunb (**a**–**c**) and *Rosa hybrida* E.H.L.Krause (**d**–**f**) under non-continuous ALAN. Different lowercase and capital letters within the columns indicate significant differences among the treatments at $p < 0.05$ in July and September, respectively. Data are mean values ± SDs ($n = 6$).

### 3.4. $CO_2$ Exchange at Night

Under continuous ALAN, the Pn of *Euonymus japonicus* Thunb at 21:00 and 2:00, subjected to a high light intensity in July and September, exhibited a significantly higher values compared to the control group; however, the Pn was unaffected by low and medium light intensities (Figure 7a,b). When exposed to non-continuous ALAN, *Euonymus japonicus* Thunb demonstrated a significant increase in Pn at 21:00 under a high light intensity in both July and September, while no significant difference was observed at 2:00 (Figure 8a,b). The Pn of *Rosa hybrida* E.H.L.Krause at 21:00 and 2:00 had no significant difference in all treatments under continuous ALAN and non-continuous ALAN, respectively, in both July and September (Figure 7c,d and Figure 8c,d).

### 3.5. Interaction between Light Intensity and Season

The interaction between light intensity and season exhibited no significant influence on the $Pn_{max}$, $A_{max}$, and Rd of both *Euonymus japonicus* Thunb and *Rosa hybrida* E.H.L.Krause, irrespective of whether the plants were subjected to continuous or non-continuous ALAN (Tables 1 and 2).

**Table 1.** The interaction of light intensity and season on the index of two plants under continuous ALAN.

| Species | Source of Variation | Pn$_{max}$ | A$_{max}$ | Rd |
|---|---|---|---|---|
| *Euonymus japonicus* Thunb | ALAN * season | ns | ns | ns |
| *Rosa hybrida* E.H.L.Krause | ALAN * season | ns | ns | ns |

ns indicates the absence of a significant relationship at $p < 0.05$.

**Table 2.** The interaction of light intensity and season on the index of two plants under non-continuous ALAN.

| Species | Source of Variation | Pn$_{max}$ | A$_{max}$ | Rd |
|---|---|---|---|---|
| *Euonymus japonicus* Thunb | ALAN * season | ns | ns | ns |
| *Rosa hybrida* E.H.L.Krause | ALAN * season | ns | ns | ns |

ns indicates the absence of a significant relationship at $p < 0.05$.

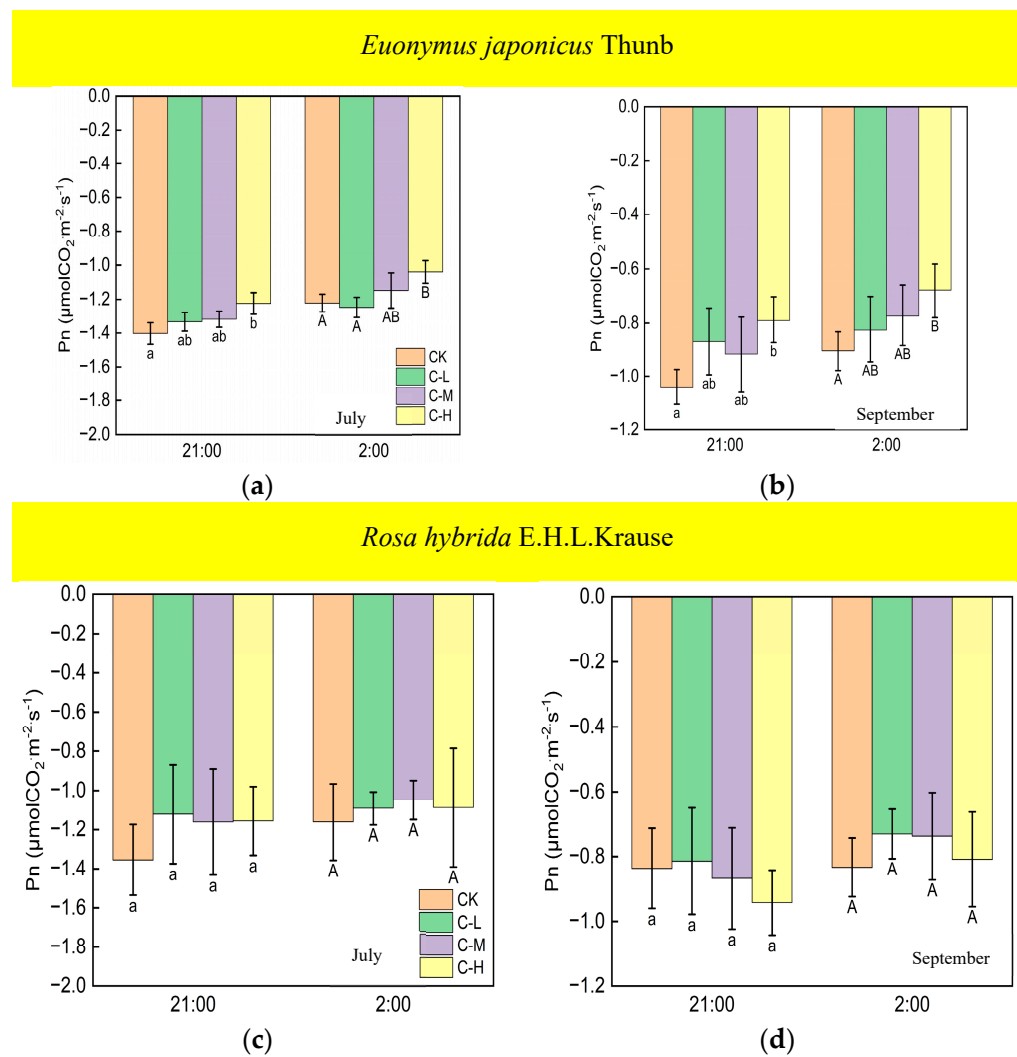

**Figure 7.** Effects of light intensity on the Pn at night of *Euonymus japonicus* Thunb (**a**,**b**) and *Rosa hybrida* E.H.L.Krause (**c**,**d**) under continuous ALAN. Different lowercase and capital letters within the columns indicate significant differences among the light treatments at $p < 0.05$ at 21:00 and 2:00, respectively. Data are mean values ± SDs ($n = 3$).

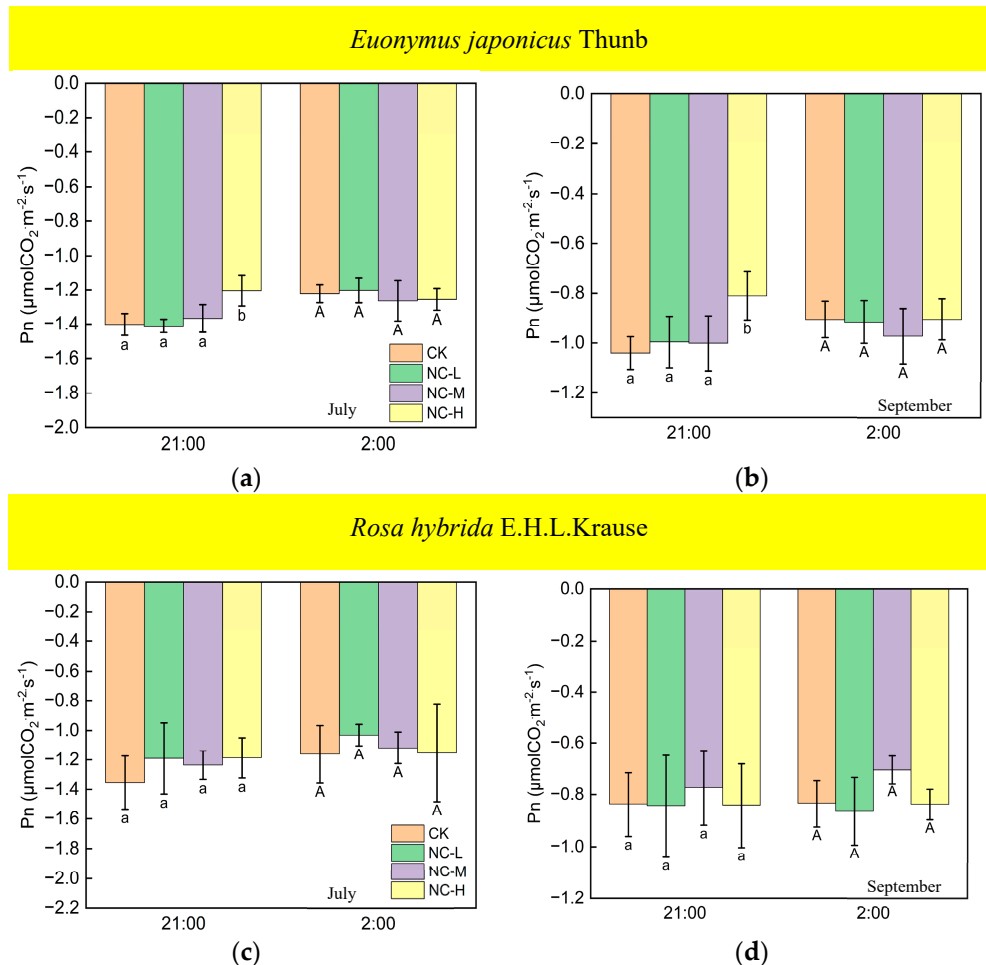

**Figure 8.** Effects of light intensity on the Pn at night of *Euonymus japonicus* Thunb (**a**,**b**) and *Rosa hybrida* E.H.L.Krause (**c**,**d**) under non-continuous ALAN. Different lowercase and capital letters within the columns indicate significant differences among the light treatments at $p < 0.05$ at 21:00 and 2:00, respectively. Data are mean values $\pm$ SDs ($n = 3$).

## 4. Discussion

### 4.1. Effects of Continuous ALAN on Photosynthesis and Respiration

In our study, we observed a significant decrease in the $Pn_{max}$, $A_{max}$, $Vc_{max}$, $J_{max}$, and TPU of *Euonymus japonicus* Thunb under low, medium, and high light intensities in continuous ALAN conditions (Figures 1a and 3a,e–g). Similarly, *Rosa hybrida* E.H.L.Krause exhibited a significant decrease in these indicators under a high light intensity in the continuous ALAN condition (Figures 1e and 3h,l–n). $Pn_{max}$ and $A_{max}$ represent the maximum photosynthetic capacity of leaves [26]. The observed decrease in the $Pn_{max}$ and $A_{max}$ of *Euonymus japonicus* Thunb and *Rosa hybrida* E.H.L.Krause under continuous ALAN indicated that continuous ALAN inhibited the photosynthesis of plants involved. $Vc_{max}$ is an important parameter that characterizes the photosynthetic capacity of plants, affecting the maximum photosynthetic rate, photorespiration, and mitochondrial respiration processes [26,27]. In our study, both *Euonymus japonicus* Thunb and *Rosa hybrida* E.H.L.Krause showed a significant decrease in $Vc_{max}$ in the continuous ALAN condition, suggesting a reduction in photosynthetic enzyme activity in these two plant species. $J_{max}$ reflects the maximum electron-transport capacity, the decrease of which in our study means that the functional balance of the electron-transport energy of the leaf is disrupted. The $J_{max}$ of plants is related to the regenerative capacity of RuBP (Ribulose-1,5- bisphosphate), the reduction of which will limit the NADPH regeneration and photophosphorylation, and inhibit the $Vc_{max}$ of plants, thus reducing their photosynthetic capacity. Triose phosphates

are an important product of the primary stage of carbon assimilation and can also be served as the regenerated material for RuBP in the regeneration stage [26,28,29]. The TPU of *Euonymus japonicus* Thunb and *Rosa hybrida* E.H.L.Krause decreased under continuous ALAN, indicating that the regenerative capacity of RuBP was reduced, which further affected the photosynthetic capacity [29]. Investigating the fluorescence properties of chlorophyll molecules helps us to understand the efficiency of intermolecular energy transfer, and fluorescence measurement is a non-destructive approach widely used to study photosynthesis [30]. In our study, we also observed a significant decrease in the ETR, $\Phi_{PSII}$, and $q_p$, of *Euonymus japonicus* Thunb under low, medium, and high light intensities in the continuous ALAN condition (Figure 5a–c). And, *Rosa hybrida* E.H.L.Krause exhibited a significant decrease in these indicators under a high light intensity in the continuous ALAN condition (Figure 5d–f). These findings indicate that continuous light stress reduces light-energy-conversion efficiency in leaves and dissipates excess light energy through increased heat dissipation [31]. Furthermore, we observed that the lc of *Euonymus japonicus* Thunb and *Rosa hybrida* E.H.L.Krause increased under a high light intensity in September (Figure 1c,g), indicating a lower light-utilization ability, which is not conducive to organic matter accumulation.

When photosynthetic organisms are exposed to stressful environmental conditions, including ALAN, the activity of photosystem II can be inhibited, resulting in a phenomenon known as photooxidation damage [16,32,33]. Stressful conditions can restrain electron transport, ATP, and NADPH synthesis, as well as the catalytic activity of Rubisco, leading to the accumulation of excessive reactive oxygen species (ROS) [33,34]. These ROS, in turn, accelerate the photodamage to photosystem II and hinder its repair. ALAN, which disrupts the natural cycle of day and night, has emerged as a novel environmental pressure. Related studies have shown that continuous light can reduce chlorophyll synthesis, $\Phi_{PSII}$, and ETR, ultimately leading to decreased photosynthesis in plants, which aligns with the findings of our study [11,16,17,19]. It has been suggested that periods of darkness are critical for various processes involved in the repair and recovery of physiological functions in many species, underscoring the importance of darkness as a resource for physiological activity [1,9]. However, exposure to ALAN disrupts the growth rhythm of plants, which typically coordinates plant physiological processes to specific times of the day or night. This disruption results in the generation of ROS, which in turn inhibits the repair of the photodamaged PSII primarily by suppressing protein synthesis [16,33]. It is worth adding that the change in the circadian clock can influence nonstructural carbohydrates' metabolism [16], leading to plants exposed to continuous light exhibiting substantial increases in nonstructural carbohydrates concentrations [16,35], which could downregulate their photosynthetic rates [15,36]. Altered biorhythms can also induce oxidative stress, impacting chlorophyll synthesis and ultimately reducing light absorption and photosynthetic efficiency [17]. Additionally, plants exposed to continuous light can influence stomatal behavior, leading to a decreased stomatal width, length, and size [37]. These changes in stomatal parameters can reduce the stomatal conductance of plants, resulting in a decreased photosynthetic rate.

Dark respiration, which consumes 30% to 80% of the total primary production of plants [38], is one of the main physiological processes in plants, and is influenced by various environmental factors, including light conditions [20]. Mitochondrial respiration under light, which is interconnected with photosynthesis, Rubisco activity, and carbon metabolism, is a complex process [20]. In our study, we observed a significant decrease in the Rd of plants under continuous ALAN (Figure 1b,f), which is consistent with findings from previous research. Research conducted by Ikkonen et al. [20] demonstrated that continuous light inhibits the respiration of eggplant and pepper per unit mass. Similarly, Matsuda et al. [39] found that the Rd of tomato plants is significantly lower under continuous-light conditions compared to a 12 h photoperiod. Continuous light exposure can alter the structure of plant leaves, leading to increased thickness and leaf area index, and these anatomical and morphological changes may be the cause of changes in leaf respiration [20]. The change in

leaf respiration may also be related to leaf nitrogen content. Proietti et al. [40] pointed out that the nitrogen content of *Eruca vesicaria* leaves would decrease under continuous-light conditions, and many studies have found a positive correlation between plant respiration and leaf nitrogen content [20,41]. Furthermore, the activity of Rubisco has been identified as another factor influencing leaf respiration [20]. Continuous light can lead to a decreased activity of Rubisco, which is associated with a lower Rd in plants.

The Rd/Ag ratio is a significant parameter reflecting the carbon balance of plants [20]. When plants are exposed to adverse environmental conditions, the Rd/Ag ratio tends to increase, which is associated with a decrease in plant growth rate [20]. In our study, the increase in the Rd/Ag ratio of *Euonymus japonicus* Thunb and *Rosa hybrida* E.H.L.Krause under continuous ALAN (Figure 1d,h) indicates that the normal carbon-balance pattern of plants is broken, which may be a factor limiting the growth of the plant.

### 4.2. Effects of Non-Continuous ALAN on Photosynthesis and Respiration

Our results showed that non-continuous ALAN did not have a significant impact on photosynthesis and respiration during the day. Cathey and Campbell [14] demonstrated that interrupting the night light for a duration of 2–4 h helps preserve the plant's natural rhythm. Additionally, An et al. [42] pointed out that extending the photoperiod to 17 h has no effect on growth parameters and biomass in the case of the autumn blessing. These studies collectively support the crucial role of darkness in maintaining normal plant growth. Meanwhile, some studies have indicated that moderately extending photoperiods can have notable effects on the physiological processes of plants [17,43]. The variations observed could be attributed to factors such as light intensity, species composition, and test species.

### 4.3. Effects of ALAN on Pn at Night

Our study showed that the Pn value of *Euonymus japonicus* Thunb at night increased significantly when the ALAN directly shone on this plant (Figure 7a,b and Figure 8a,b). This increase could be attributed to either the decrease in respiration or the driving of the photosynthesis of *Euonymus japonicus* Thunb at night. However, since the light intensity during the night-time period was lower than the compensation point (Figure 1c and Table S1), the Pn of *Euonymus japonicus* Thunb was still negative. In future research, it is crucial to investigate and distinguish between the effects of ALAN on photosynthesis and respiration at night independently. This will allow for a more comprehensive understanding of the specific mechanisms underlying the observed changes in plant responses to ALAN exposure.

### 4.4. Difference in the Responses of Photosynthesis and Respiration between Euonymus japonicus Thunb and Rosa hybrida E.H.L.Krause under ALAN

The impact of ALAN on plants can vary, and it is influenced by the inherent morphological and physiological characteristics of the species [1,17,18,44]. In our measurements, the height of the *Rosa hybrida* E.H.L.Krause exceeded that of the *Euonymus japonicus* Thunb during our measurement, resulting in its upper portion being exposed to more intense night-time light (Table S1). We found that low and medium light intensities had significant effects on the photosynthesis and respiration of *Euonymus japonicus* Thunb in the day time. However, these light intensities did not have significant effects on *Rosa hybrida* E.H.L.Krause. This disparity may be attributed to the stronger antioxidant capacity of *Rosa hybrida* E.H.L.Krause compared to *Euonymus japonicus* Thunb (Table S2), enabling it to better mitigate the stress caused by night-time light exposure through the effective removal of ROS. Moreover, in the night, ALAN with high intensity significantly affected the Pn of *Euonymus japonicus* Thunb, but did not significantly affect the Pn of *Rosa hybrida* E.H.L.Krause. The reason for this difference may be due to the physiological differences between the two plants. To further comprehend these differences at the physiological level, future studies should delve into the underlying reasons and mechanisms associated with plant responses to ALAN stress.

### 4.5. Prospects and Shortcomings

Several studies have suggested that diurnal fluctuations in air temperature have the potential to reset circadian rhythms that have been disrupted by ALAN, and this resetting mechanism may alleviate or even eliminate the negative effects induced by continuous light exposure in certain species [36]. However, the specific physiological mechanisms underlying these effects remain unclear. It is still necessary to investigate whether temperature or other factors influence the impact of ALAN on the two plant species involved. Some studies have confirmed that ALAN will affect plant phenology and pollination, and these factors will ultimately affect the physiological process of plants [45,46]. However, our study did not consider the impact of these factors on plant photosynthesis and respiration, which is the shortcoming of our experiment. Overall, a unified hypothesis regarding the effects of ALAN on plants has not yet been proposed. The responses of plant photosynthesis and respiration to ALAN stimuli vary depending on factors such as light intensity, light duration, spectrum, plant species, and other environmental conditions. Further research is needed to explore these complex interactions and elucidate the underlying mechanisms.

It is important to acknowledge that the light intensity mentioned in this paper refers to the light intensity at the center of the plot ground. Since the plants were pruned in the winter of 2021, the night-time light intensity experienced by the plants in 2022 was not a constant value but gradually increased with the plants growing. However, the field measurements indicated that even at high night-time light intensities, the light intensity received by the uppermost part of *Euonymus japonicus* Thunb and *Rosa hybrida* E.H.L.Krause remained below their lc (Figure 1c,g and Table S1). Because the plants selected for measurement in each plot were located closest to the center of the light source, three gradients of different night light intensities can still be guaranteed. Furthermore, it is worth noting that our current study solely focused on the scale of individual plant leaves and did not consider the scale of the entire plant canopy. The ALAN will decrease from top to bottom or from the outer edges to the center of the plant canopy due to mutual shading. Further investigation is required to determine whether the leaves not directly exposed to ALAN undergo changes in physiology and morphology, thereby affecting the overall growth and the carbon balance of the plant.

## 5. Conclusions

In this study, the effects of ALAN with different light intensities on the photosynthesis and respiration of two species were investigated under continuous and non-continuous ALAN, respectively, to develop an understanding of the thresholds and dose–response relationships of light sensitivity in plants, particularly at night-time light intensities below the lc of the plant. Our findings demonstrate that continuous ALAN, across all three light intensities, inhibits the photosynthesis and respiration of *Euonymus japonicus* Thunb, thereby disrupting the carbon balance of the leaves. This inhibitory effect was most prominent at higher light intensities. By contrast, only high light intensities had similar effects on *Rosa hybrida* E.H.L.Krause, suggesting that *Rosa hybrida* E.H.L.Krause has a greater resistance to ALAN compared to *Euonymus japonicus* Thunb. Non-continuous ALAN, however, did not significantly impact the photosynthesis, respiration, and carbon-balance patterns of either of the two species during the day. Moreover, we observed that a high light intensity could also affect the Pn value of *Euonymus japonicus* Thunb during the night time. To mitigate the impact of ALAN on plants and maintain a normal carbon-balance pattern, it is effective to turn off light sources at night, reduce their light intensity and plant ALAN-resistant plants. Future studies should explore the effects of different types of artificial light sources combined with other environmental conditions on the photosynthesis and respiration processes of plants at the canopy scale.

**Supplementary Materials:** The following supporting information can be downloaded at https://www.mdpi.com/article/10.3390/f15040659/s1, Figure S1. Wavelength distributions in low (a), middle (b) and high (c) intensities respectively; Figure S2. Images of the field experiment

during the day and at night; Table S1. light intensity received by the uppermost part of *Euonymus japonicus* Thunb and *Rosa hybrida* E.H.L.Krause in different treatments (lux); Table S2. Total antioxidant capacity of *Euonymus japonicus* Thunb and *Rosa hybrida* E.H.L.Krause in July and September under no ALAN ($\mu$mol $Fe^{2+}g^{-1}$).

**Author Contributions:** Y.W.: writing—original draft, conducted experiments; J.Z. (Jiaolong Zhang): method; Z.L.: method, writing—review; D.H.: review and editing; J.Z. (Jian Zhang): method. All authors have read and agreed to the published version of the manuscript.

**Funding:** This research was supported by the National Natural Science Foundation of China (No. 42071274).

**Data Availability Statement:** Data can be made available on request.

**Conflicts of Interest:** The authors declare that they have no known competing financial interests or personal relationships that could have appeared to influence the work reported in this paper.

## Abbreviations

| | |
|---|---|
| ALAN | artificial light at night |
| $Pn_{max}$ | maximum net photosynthetic rate |
| lc | light-compensation point |
| Rd | dark-respiration rate |
| $A_{max}$ | photosynthetic capacity |
| $\Gamma$ | $CO_2$-compensation point |
| $Ci_{sat}$ | saturated intercellular $CO_2$ concentration |
| Rp | photorespiration rate |
| $Vc_{max}$ | maximum carboxylation rate |
| $J_{max}$ | maximum electron-transfer rate |
| TPU | triose phosphate-utilization rate |
| ETR | apparent electron-transfer rate |
| $\Phi_{PSII}$ | effective quantum yield |
| $q_p$ | photochemical quenching parameter |
| Pn | net photosynthetic rate |
| ROS | excessive reactive oxygen species |

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
