# Peer review of "Effects of Artificial Light at Night on Photosynthesis and Respiration of Two Urban Vascular Plants"

_forests, doi:10.3390/f15040659_

Round 1

Reviewer 1 Report

Comments and Suggestions for Authors

The study evaluated the effect of illumination of plant during night time on the primary metabolic processes such as photosynthesis and respiration of two plant species. The authors concluded that continues lighting during nights inhibited photosynthesis and respiration of E. japonicus, but R. hybrida, had a greater resistance to this factor than E. japonicus.

Comments:

Lines 519-531: Please, use italic for the species;

Line 16: Remove (E. japonicus) and (R. hybrida), indicate the author of the species;

Line 47: What ‘cave systems’ means?

Line 57: extra point;

Line 62-64: The sentences needs correction. Don’t use ‘dark role’ or ‘dark refuges’. Maybe ‘ the role of darkness’?

Line 81: Don’t use italic;

Line 95-96: Add references, please;

Line 103; Use total species name and author here;

Line 157, 163: Some corrections needed; Use uniform for ‘μmolm-2s -1’ throughout the text;

Line 148,179,188: What ‘For either species’ means? Eight?

All Figures are of low quality and must be improved;

In Figures titles ‘a, b, c, and so on should be indicated;

Please reduce all % to the nearest whole number;

In the section Results, it is difficult for readers to understand what plant species and light regime the information relates to. Clarify, please, throughout the text of the section;

Line 293: It seems to be better to use ‘CO2 exchange’ instead of “Pn” in the night;

Line 414-416: What Figure illustrate this statement?

Author Response

Reviewer 1 Comments:

The study evaluated the effect of illumination of plant during night time on the primary metabolic processes such as photosynthesis and respiration of two plant species. The authors concluded that continues lighting during nights inhibited photosynthesis and respiration of E. japonicus, but R. hybrida, had a greater resistance to this factor than E. japonicus.

Response: We would like to thank you for your constructive comments concerning our article. These comments and suggestions are all valuable and helpful for improving our article. According to your comments, we have tried best to modify our manuscript to meet your approval. The responses to your comments are presented following:

Lines 519-531: Please, use italic for the species;

Response: Thanks for your review. In the new version the name of the species has been italicized. (line 707, 710,711,722,760,782)

Line 16: Remove (E. japonicus) and (R. hybrida), indicate the author of the species;

Response: Thanks for your review. In the new version, (E. japonicus) and (R. hybrida) have been removed, and the author of the species have been indicated. (line 17)

Line 47: What ‘cave systems’ means ?

Response: Thanks for your review. A cavern system is an interconnected cavern pipe, such as a karst cave, accessible by man or hydraulically connected to a specific area. Cave systems are unique ecosystems with weak light, constant temperature, high humidity, and poor nutrition.

Line 57: extra point;

Response: Thanks for your review. In the new version the extra point has been removed. (line 71)

Line 62-64: The sentences needs correction. Don’t use ‘dark role’ or ‘dark refuges’. Maybe ‘ the role of darkness’?

Response: Thanks for your review, and this is a very good suggestion. In the new version, dark role has been changed to ‘the role of darkness’. (line 79)

Line 81: Don’t use italic;

Response: Thanks for your review. In the new version we corrected it. (line 102)

Line 95-96: Add references, please;

Response: Thanks for your review. In the new version the references were added (line 121): 

Line 103; Use total species name and author here;

Response: Thanks for your review. In the new version the total species name and author have been used. (line 130,131)

Line 157, 163: Some corrections needed; Use uniform for ‘μmolm-2s -1’ throughout the text;

Response: Thanks for your review. In the process of measuring net photosynthetic rate (Pn) responses to photosynthetic photon flux density (PPFD) and CO2 concentration, the unit of light intensity is μmolm-2s-1, and the unit of carbon dioxide concentration is μmol mol-1. In the text, we unify the two units respectively.

Line 148,179,188: What ‘For either species’ means? Eight?

Response: Thanks for your review. We were sorry for this confusing expression. What we really wanted to state is ‘either of two species’. And we have changed ‘either species’ to  ‘either of two species’ in the revised version. (line 193,233,244,658,659,)

All Figures are of low quality and must be improved;

Response: Thank you very much for pointing out the shortcomings of our figures. In the new version we redrew all the figures to improve the quality of the paper.

In Figures titles ‘a, b, c, and so on should be indicated;

Response: Thanks for your review. In the new version the ‘a, b, c, and so on have been indicated.

Please reduce all % to the nearest whole number;

Response: Thanks for your review. In the new version the % have been reduced to the nearest whole number.

In the section Results, it is difficult for readers to understand what plant species and light regime the information relates to. Clarify, please, throughout the text of the section;

Response: Thanks for your review, and that is a very good suggestion. In the new version, we annotated the figure corresponding to information about plant species and light intensity, so that readers can better understand what plant species and light regime the information relates to. (line 451,452,454,483,485,486,490,491,531,532,554,573,)

Line 293: It seems to be better to use ‘CO2 exchange’ instead of “Pn” in the night;

Response: Thanks for your review. In the new version the Pn have been replaced by CO2 exchange. (line 243,402)

Line 414-416: What Figure illustrate this statement?

Response: Thanks for your review.  Figure 7 (a,b) and Figure 8 (a,b) revealed a significant increase in the Pn of Euonymus japonicus Thunb. when the high light intensity is directly illuminated. Since Pn is negative at night, the smaller its absolute value is, the larger it is. In the new version, we label the source of the content. (line 571-573)

Reviewer 2 Report

Comments and Suggestions for Authors

This is my first review of the manuscript entitled "Effects of artificial light at night on photosynthesis and respiration of two urban vascular plants" by Wei et al. (V2)

The manuscript focuses on the light pollution / artificial light at night effects on plant performance in photosynthesis and respiration. The research topic is timely, and the manuscript under review adds up some novel data acquired using two species in a field experiment.

The mansucript is written in good English, is easily read and requires only a few subtle editings. My comments below are targeted to further improve the soundness of the manuscript and to avoid some potentially ambigous statements in the text.

Reader may guess the two species selected by the authors as a model plants (Euonymus japonicus and Rosa hybrida) are widely used or grown for urban greening in China. However, it is better to explain the reasons for such choice. Also, the choice of seasons for measurements (July and September) should be substantiated (first appeared on L192).

Consequently, on L247 authors say 'interaction between night light intensity and month', however, this hypothetical interaction is better understood as an interaction betweet light intensity and seasonal changes/state of plants.

L190–191: It is unclear what type of air flow rate is referred to here, and why its measurement units are umol/s.

L237 etc.: qp – photochemical quenching – for accepted symbols, authors are welcome to refer to the paper Goltsev, V.N., Kalaji, H.M., Paunov, M. et al. Variable chlorophyll fluorescence and its use for assessing physiological condition of plant photosynthetic apparatus. Russ J Plant Physiol 63, 869–893 (2016). https://doi.org/10.1134/S1021443716050058

Figures are fine, however increasing font size would be a good idea, and the captions should be supplemented with sample size information (n=?)

Figure 1 (L290) lacks the 'Eounymus japonicus' row above the subfigures 1a,b,c,d.

Figure S1 should be supplemented with Y axis title (relative intensity at the wavelength). Currently it makes the wrong impression that the total powers of all three light intensities were equal.

For the convenience of reader, I would suggest to move abbreviations list to the very beginning of the manuscript (right before the Introduction, please negotiate this with the scientific editor or the typesetter). Also, reminder of the abbreviation on its forst appear in Results and Discussion sections will definitely help the reader.

Authors should correct the misleading typo 'TUP' on L411.

On the same page at L429-430, ribulose-1,5-bisphosphate is mistakenly referred to as 'diphosphate'. This is incorrect: diphosphate means two phosphate groups are consequently bound to the same carbon atom in the molecule, whereas bisphosphate means two phosphate groups bound to two different carbon atoms, which is the case for RuBP.

In my opinion, the statement that activity of PSII can be photoinhibited under ALAN, is an overestimation. Light intensities of ALAN are far from saturating light intensity that causes photoinhibition. The statement that ALAN could be joined to other environmental stress could be doubtful, because, again, its intensity is much lower than in the mid-day, and it is not generally combined with other types of simultaneous environmental factors such as drought and/or high temperature. Please comment on that.

Other than that, authors have prepared the solid manuscript which conclusions are supported with data, and their shortcomings are also mentioned. To sum up, the manuscript would be of an interest to the general reader and for the design of further studies.

Comments on the Quality of English Language

L157: may be better to spell 'spectral distribution' instead of 'spectral composition'.

L186: terminology: the LI-COR device LI-6400XT is the photosynthesis analysis system, not the 'photosynthesis system; itself.

L200, L203 etc.: missed dot symbol in 'μmolm-2s-1'.

L216 etc.: logically, the symbol used for 'light compensation point' should be lc (lowercase letters L and C), however in manuscript it looks like capital I and lowercase C. Please check hereinafter.

L532–533: please check the sentence, it looks incomplete.

Author Response

Comments and Suggestions for Authors

This is my first review of the manuscript entitled "Effects of artificial light at night on photosynthesis and respiration of two urban vascular plants" by Wei et al. (V2)

The manuscript focuses on the light pollution / artificial light at night effects on plant performance in photosynthesis and respiration. The research topic is timely, and the manuscript under review adds up some novel data acquired using two species in a field experiment.

The mansucript is written in good English, is easily read and requires only a few subtle editings. My comments below are targeted to further improve the soundness of the manuscript and to avoid some potentially ambigous statements in the text.

Response: Thank you for your comments concerning our manuscript. Those comments are valuable and very helpful. We have read through comments carefully and have made corrections. Based on the instructions provided in your letter, we uploaded the file of the revised manuscript. The responses to your comments are presented following:

Reader may guess the two species selected by the authors as a model plants (Euonymus japonicus and Rosa hybrida) are widely used or grown for urban greening in China. However, it is better to explain the reasons for such choice. Also, the choice of seasons for measurements (July and September) should be substantiated (first appeared on L192).

Response: Thanks for your review, and this is a very good suggestion. As you mentioned Euonymus japonicus and Rosa hybrida are widely used or grown for urban greening in China. In the new version, relevant content has been added in line 130-132.

Because July is the flourishing period of the growth of the two plants, and September is the beginning of the decline period of the growth of the two plants, we chose July and September to measure the relevant indicators of the two plants. In the new version, relevant content has been added in line 182-187.

Consequently, on L247 authors say 'interaction between night light intensity and month', however, this hypothetical interaction is better understood as an interaction betweet light intensity and seasonal changes/state of plants.

Response: Thanks for your review, and this is a very good suggestion. In this study, July represents the typical summer time and September represents the typical autumn time. The influence of the interaction of light intensity and season on the related indicators was analyzed by two-factor ANOVA. In the new version “Interaction between light intensity and month” has been changed to “Interaction between light intensity and season”. (line 252,256,434,435 440 and 445)

L190–191: It is unclear what type of air flow rate is referred to here, and why its measurement units are umol/s.

Response: Thanks for your good comment. This may be a complex physical problem with the measuring device that is beyond our expertise. In the process of photosynthesis measurement with LI-6400, the air velocity is usually expressed in umol/s. Examples include the following:

(1) Zhang, L.L., Zhang, S., 2019. The quantitative impact of different leaf temperature determination on computed values of stomatal conductance and internal CO2 concentrations. Agric. For. Meteorol. 279, 107700 https://doi.org/10.1016/j. agrformet.2019.107700.

(2) Zou, J., Fanourakis, D., Tsaniklidis, G., Cheng, R., Yang, Q., and Li, T. (2021). Lettuce growth, morphology and critical leaf trait responses to far-red light during cultivation are low fluence and obey the reciprocity law. Sci. Hortic. 289, 110455. doi: 10.1016/j.scienta.2021.110455

(3) Tominaga J., Kawamitsu Y.: Cuticle affects calculations of internal CO2 in leaves closing their stomata. – Plant Cell Physiol. 56: 1900-1908, 2015. DOI10.1093/pcp/pcv109

L237 etc.: qp – photochemical quenching – for accepted symbols, authors are welcome to refer to the paper Goltsev, V.N., Kalaji, H.M., Paunov, M. et al. Variable chlorophyll fluorescence and its use for assessing physiological condition of plant photosynthetic apparatus. Russ J Plant Physiol 63, 869–893 (2016). https://doi.org/10.1134/S1021443716050058

Response: We are very grateful to the reviewer for this good comment and the literature suggested.  In the new version, “qp”has been changed to “qp”. (line 242,370,376,379,384,386,388,393,398,481 681,Fig5 c f, Fig6 c f)

Figures are fine, however increasing font size would be a good idea, and the captions should be supplemented with sample size information (n=?)

Response: Thanks for your good comment. Considering that we are all reading literature on the Internet now, we can enlarge the picture, so we did not enlarged the font. The sample size information has been added to the captions. (line 312,330,363,368,396,401,419,433)

Figure 1 (L290) lacks the 'Eounymus japonicus' row above the subfigures 1a,b,c,d.

Response: Thanks for your review. the 'Eounymus japonicus' row has been added above the subfigures 1a,b,c,d. (line 307,308 )

Figure S1 should be supplemented with Y axis title (relative intensity at the wavelength). Currently it makes the wrong impression that the total powers of all three light intensities were equal.

Response: Thank you very much for pointing out the shortcomings of the Figure S1. The unit of the ordinate is (μW/cm2)/nm, which represents the intensity of the light. The unit of light intensity used in our study was lux. In order to avoid ambiguity, we did not show the unit of ordinate during the measurement. This is indeed a mistake and deficiency in our work. In fact, because of the different ordinate values, the energy of the three light intensities is different. Since this indicator was measured in the sample plot in 2021 and it has now expired and been dismantled, we could not make the supplementary measurement. However, Figure S1 could still convey the most important information. Specifically, it showed the spectral distribution of each band and indicated the spectral distribution is roughly the same under the three light intensities. In the following research, we will pay attention to the problems you mentioned and improve the measurement method.

For the convenience of reader, I would suggest to move abbreviations list to the very beginning of the manuscript (right before the Introduction, please negotiate this with the scientific editor or the typesetter). Also, reminder of the abbreviation on its forst appear in Results and Discussion sections will definitely help the reader.

Response: Thanks for your review. We will contact the editor to determine the location of the abbreviation list according to the journal's requirements. And we further checked and corrected the abbreviations in the discussion and result. (line 509,554,632,647,650)

Authors should correct the misleading typo 'TUP' on L411.

Response: Thanks for your review. In the new version, TUP has been changed to “TPU”. (line 450)

On the same page at L429-430, ribulose-1,5-bisphosphate is mistakenly referred to as 'diphosphate'. This is incorrect: diphosphate means two phosphate groups are consequently bound to the same carbon atom in the molecule, whereas bisphosphate means two phosphate groups bound to two different carbon atoms, which is the case for RuBP.

Response: Thank you for pointing out our mistake. In the new version, ribulose 1, 5-diphosphate  has been changed to Ribulose-1,5-bisphosphate. (line 468,469)

In my opinion, the statement that activity of PSII can be photoinhibited under ALAN, is an overestimation. Light intensities of ALAN are far from saturating light intensity that causes photoinhibition. The statement that ALAN could be joined to other environmental stress could be doubtful, because, again, its intensity is much lower than in the mid-day, and it is not generally combined with other types of simultaneous environmental factors such as drought and/or high temperature. Please comment on that.

Response: We are very grateful to the reviewer for this good comment.  Compared with the light intensity in the daytime, the intensity of artificial light at night is far lower than the light saturation point of plants and cannot directly cause photoinhibition on plants. However, continuous artificial light at night will break the growth rhythm of plants, produce more reactive oxygen species in plants or cause the accumulation of carbohydrates in plants, and damage the normal function of photosystem II, which can indirectly cause daytime photoinhibition in plants. Furthermore, we consider that there is an ambiguity in the use of photoinhibition here, so we replace photoinhibition with photooxidation damage by referring to the study of Velez-Ramirez(2011). (Velez-Ramirez, A.I., van Ieperen, W., Vreugdenhil, D., Millenaar, F.F., 2011. Plants under continuous light. Trends Plant Sci 16, 310-318. DOI10.1016/j.tplants.2011.02.003). (line 496)

Other than that, authors have prepared the solid manuscript which conclusions are supported with data, and their shortcomings are also mentioned. To sum up, the manuscript would be of an interest to the general reader and for the design of further studies.

Comments on the Quality of English Language

L157: may be better to spell 'spectral distribution' instead of 'spectral composition'.

Response: Thanks for your review. In the new version, 'spectral composition' has been changed to “spectral distribution”. (line 157)

L186: terminology: the LI-COR device LI-6400XT is the photosynthesis analysis system, not the 'photosynthesis system; itself.

Response: Thanks for your review. In the new version, photosynthesis system has been changed to photosynthesis analysis system. (line 191)

L200, L203 etc.: missed dot symbol in 'μmolm-2s-1'.

Response: Thanks for your review, and that is a very good suggestion. We refer to the relevant literature and add a space between “μmol” and “m-2s-1”. (line 205,208,239). The references are as follows:

(1) Zhang, L.L., Zhang, S., 2019. The quantitative impact of different leaf temperature determination on computed values of stomatal conductance and internal CO2 concentrations. Agric. For. Meteorol. 279, 107700 https://doi.org/10.1016/j. agrformet.2019.107700.

(2) Zou, J., Fanourakis, D., Tsaniklidis, G., Cheng, R., Yang, Q., and Li, T. (2021). Lettuce growth, morphology and critical leaf trait responses to far-red light during cultivation are low fluence and obey the reciprocity law. Sci. Hortic. 289, 110455. doi: 10.1016/j.scienta.2021.110455

(3) Tominaga J., Kawamitsu Y.: Cuticle affects calculations of internal CO2 in leaves closing their stomata. – Plant Cell Physiol. 56: 1900-1908, 2015. DOI10.1093/pcp/pcv109

L216 etc.: logically, the symbol used for 'light compensation point' should be lc (lowercase letters L and C), however in manuscript it looks like capital I and lowercase C. Please check hereinafter.l

Response: Thanks for your review. In the new version, the symbol used for light compensation point has been changed to lc (lowercase letters L and C). (line 221,289,290,292,309,327,489,681)

L532–533: please check the sentence, it looks incomplete.

Response: Thank you for your comment. In the new version, the sentence “Our study revealed a significant increase in the Pn of Euonymus japonicus Thunb when the high light intensity is directly illuminated” was changed to “Our study showed that the Pn value of Euonymus japonicus Thunb at night increased significantly when the ALAN directly shone on this plant (line 571-573)

Reviewer 3 Report

Comments and Suggestions for Authors

Comments to Wei et al. “Effects of artificial light…” (ms 2919892)

Supplementary:

                                 Figure S1. Axes should be properly labelled; horizontal add “nm”; vertical, make clear if scale refers to “energy per wavelength interval, relative” or something else, that would affect the shape of the graphs (e.g., photons per wavelength interval). “Wavelength” is usually written as one word.

                                 “Note:50 lux≈1μmoles photons m-2s-1” Adjust spaces here and elsewhere.

                                 Lux is a unit of illuminance, not of light intensity.

Manuscript

                                 Check how your use of “intensity” agrees with, e.g., https://en.wikipedia.org/wiki/Light_intensity

Line

19                          Is “Our findings indicate that” necessary. There are perhaps other places          where the language can be less wordy without loss of content.

54                          “most essential”: In my view something is either essential or not essential.

56                          “the photosynthesis”  “photosynthesis”

                                 It would have been interesting to know something about    the natural light during the experimental time, at least the times for sunset and sunrise.

Comments on the Quality of English Language

Can be improved; see above.

Author Response

Thank you for spending time in reviewing our manuscript and providing us with a list of constructive comments and suggestions. We have carefully considered all your comments and have made changes and explanations accordingly. The responses to your comments are presented following:

Supplementary:

Figure S1. Axes should be properly labelled; horizontal add “nm”; vertical, make clear if scale refers to “energy per wavelength interval, relative” or something else, that would affect the shape of the graphs (e.g., photons per wavelength interval). “Wavelength” is usually written as one word.

Response: Thank you very much for pointing out the shortcomings of the Figure S1. In the new version, “nm” has been added to horizontal axis. The unit of the ordinate is (μW/cm2)/nm, which represents the intensity of the light. But the unit of light intensity used in our study was lux. In order to avoid ambiguity, we did not show the unit of ordinate during the measurement and in the last manuscript. This is indeed a mistake and deficiency in our work. Since this indicator was measured in the sample plot in 2021 and it has now expired and been dismantled, we could not make the supplementary measurement. However, Figure S1 could still convey the most important information. Specifically, it showed the spectral distribution of each band and indicated the spectral distribution is roughly the same under the three light intensities. In the following research, we will pay attention to the problems you mentioned and improve the measurement method.  And in the new version, “Wave length” has been changed to “Wavelength”

“Note:50 lux≈1μmoles photons m-2s-1” Adjust spaces here and elsewhere.

Response: Thanks for your review. This is a constructive question. The relationship between “lux” and “μmoles photons m-2s-1” is affected by the spectral distribution and color temperature of the light source. In the study, our field measurements show that 50 lux≈1μmoles photons m-2s-1 in an artificial light source with a white spectrum and 6500 color temperature.

Lux is a unit of illuminance, not of light intensity.

Response: Thanks for this good question. The light intensity of artificial light at night could also generally measured by lux, and lux is commonly used in similar studies on artificial light at night research, such as:

  1. Hey M H, DiBiase E, Roach D A, et al. Interactions between artificial light at night, soil moisture, and plant density affect the growth of a perennial wildflower[J]. Oecologia, 2020, 193, 503-510.
  2. Bennie J, Davies T W, Cruse D, et al. Artificial light at night alters grassland vegetation species composition and phenology[J]. Journal of Applied Ecology, 2018, 55, 442-450.
  3. Speisser B, Liu Y J, van Kleunen M. Biomass responses of widely and less-widely naturalized alien plants to artificial light at night[J]. Journal of Ecology, 2021, 109, 1819-1827.

Thus, in this study we only employed “lux” to measure the intensity of artificial light at night. Of course, ‘μmol photons m-2 s-1 ’is an another important and reasonable light unit when studying the plant photosynthesis and respiration response to natural light. In the following study, we will further clarify the relationship between “lux” and “μmol photons m-2 s-1” and use “μmol photons m-2 s-1”.

Manuscript

Check how your use of “intensity” agrees with, e.g., https://en.wikipedia.org/wiki/Light_intensity

Response: Thank you very much for your question and URL. We are very sorry that we cannot open the website you provided in our country, so we can't get the relevant information in time. Three light intensities, 22 lux,57 lux and 94 lux, were used in our study, which might not meet some criteria for the definition of light intensity levels. However, the three light intensities of 22 lux, 57 lux and 94 lux used in our study established a light gradient, i.g. low, medium and high light intensities.

Line19 Is “Our findings indicate that” necessary. There are perhaps other places where the language can be less wordy without loss of content.

Response: Thanks for your review. In the new version, we have polished the language to make the article less wordy and more concise. (line 19-25)

54 “most essential”: In my view something is either essential or not essential.

Response: Thanks for your review. In the new version, we deleted the word “most” to make the expression of the sentence more reasonable. (line 53)

56   “the photosynthesis”  “photosynthesis”

Response: Thanks for your review. In the new version “the photosynthesis” has been changed to “photosynthesis”. (line 55)

It would have been interesting to know something about the natural light during the experimental time, at least the times for sunset and sunrise.

Response: Thanks for your review. This is a constructive question. During the whole test period, the light intensity during the daytime was natural light intensity and was not affected by other human factors. We turned on all lights at dusk and turned off some before dawn. The light source was turned on and off at different times depending on the time of sunset and sunrise. We did not measure the intensity of natural light, which is really a shortcoming of our study. In C-L, C-M and C-H treatments, the plants were in continuous light conditions throughout the experiment and did not experience darkness. In NC-L, NC-M and NC-H treatments, the plants can get darkness for several hours after the light source was turned off at 23:00. In the following related study, we will measure the natural light during the day, especially at dusk and dawn.

Round 2

Reviewer 1 Report

Comments and Suggestions for Authors

The manuscript was improved according the comments. But one comment more:

Authors should use the full species name and author only once when first mentioning the species. Further in the text, the short name of the species should be used without indicating the author. Don't use italic for the author's name.

First mention - Euonymus japonicus Thunb

Second mention – E. japonicus